# Continual Hyperbolic Learning of Instances and Classes

## Abstract

Instance-level continual learning addresses the challenging problem of recognizing and remembering specific instances of object classes in an incremental setup, where new instances appear over time. Continual learning of instances forms a more fine-grained challenge than conventional continual learning, which is only concerned with incremental discrimination at the class level. In this paper, we argue that for real-world continual understanding, we need to recognize samples both at the instance and class level. We find that classes and instances form a hierarchical structure and propose HyperCLIC, a hyperbolic continual learning algorithm for visual instances and classes, to enable us to learn from this structure. We introduce continual hyperbolic classification and distillation objectives, allowing us to embed the hierarchical relations between classes and from classes to instances. Empirical evaluations show that HyperCLIC can operate effectively at both levels of granularity and with better hierarchical generalization, outperforming well-known continual learning algorithms. The code is included with this submission and will be made publicly available.

## 1 Introduction

Continual learning addresses a long-standing challenge in machine learning: learning from new classes leads to the catastrophic forgetting of old classes (Kirkpatrick et al., 2017; Wu et al., 2019). As a result of catastrophic forgetting, we cannot simply fine-tune models on new data. Many solutions have been proposed to enable incremental learning from new classes while retaining recognition performance from old classes. Well-known solutions include data replay (Bang et al., 2021; Wang et al., 2021), model regularization (Yin et al., 2021; Lee et al., 2020), and knowledge distillation (Kang et al., 2022; Dong et al., 2022).

Where continual learning typically performs class-level discrimination, recent works have broadened the scope to instance-level continual learning. Instance-level continual understanding is essential in many real-world domains such as robotics, where classifying the specific instance of objects helps the robot to decide where to place or how to use them (Ammirato, 2019; Singh et al., 2014; Held et al., 2016). The EgoObjects dataset (Zhu et al., 2023) and its latest instance-level continual challenge (Pellegrini et al., 2022) highlight the difficult and open-ended nature of this problem.

In this work, we strive to perform continual learning simultaneously at the instance and class level. In many real-world domains, recognition only at one granularity level is insufficient. On one hand, class-level continual learning is invariant to class instances by design. On the other hand, instance-level continual learning without class-level awareness is prone to making significant mistakes. For domains such as robotics, self-driving cars, and more, it is crucial that in the case of instance-level mistakes, the class-level prediction is still correct to avoid accidents and to be able to generalize to new instances quickly. Classes and instances are hierarchically related, as classes have their coarse-to-fine hierarchy (Fellbaum, 2010; Deng et al., 2009), and instances add an additional layer to the hierarchy (Yan et al., 2024). Figure 1 visualizes the hierarchical organization of classes and their instances.

We propose HyperCLIC, a hyperbolic continual learner that leverages the class-instance hierarchy for joint instance- and class-level recognition. We first show how the class-instance hierarchy can be embedded as prototypes in hyperbolic space. We then outline hyperbolic classification and distillation losses to enable incremental learning and hierarchical knowledge retention at both levels of granularity. Experimental evaluation on EgoObjects, Split CIFAR-100 and CORe50 highlight the potential of our approach, with the ability to perform high-quality recognition at both instance- and class-level.

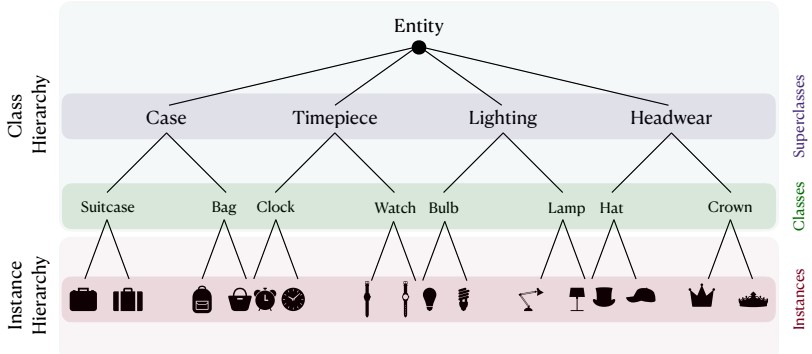

Figure 1: **Recognizing classes and other instances simultaneously** is important in many real-world applications. By adding instances as an additional layer to the object hierarchy and learning representations that capture the joint hierarchy, we can classify samples at multiple levels of granularity.

## 2 Literature Review

### 2.1 Continual Learning

Continual learning focuses on the ability of a model to learn and retain knowledge incrementally over time. This field is studied under three main scenarios (Van de Ven et al., 2022; Chen & Liu, 2018): Task-incremental, class-incremental, and domain-incremental learning. In task-incremental learning (Van de Ven et al., 2022; Kirkpatrick et al., 2017; Li & Hoiem, 2017), a model is trained on incremental tasks with clear boundaries. The task ID is known during test time. In contrast, in the class-incremental learning (Guo et al., 2022a; Kim et al., 2022; 2023; Wu et al., 2019; Rebuffi et al., 2017), the task ID is not provided, making task-incremental a particular case of class-incremental. Domain-incremental (Shi & Wang, 2024; Garg et al., 2022b; Kalb et al., 2021; Mirza et al., 2022; Wang et al., 2022b) focuses on scenarios where the data distribution's incremental shift is explicitly modeled. In this work, we will concentrate on class-incremental learning.

Regardless of continual scenarios, the main challenge of continual learning is to alleviate catastrophic forgetting with only limited access to the previous data (Evron et al., 2022; Shi et al., 2021). Catastrophic forgetting means that performance on previously learned tasks degrades significantly when learning new tasks. To address this, various strategies have been developed. Zhou et al. (2023) groups class-incremental methods into data-centric, model-centric, and algorithm-centric. Data-centric methods concentrate on solving class-incremental learning with exemplars by using data replay (Prabhu et al., 2020; Bang et al., 2021; Chaudhry et al., 2018a; Aljundi et al., 2019; Rolnick et al., 2019; Shin et al., 2017; Wang et al., 2021) or data regularization techniques (Lopez-Paz & Ranzato, 2017; Chaudhry et al., 2018b; Zeng et al., 2019). Model-centric methods either regularize the model parameters from drifting away (Kirkpatrick et al., 2017; Zenke et al., 2017; Yin et al., 2021; Lee et al., 2020) or dynamically expand the network structure for stronger representation ability (Yan et al., 2021; Zhou et al., 2022b; Wang et al., 2022a; Douillard et al., 2022). Algorithm-centric methods either utilize knowledge distillation to resist forgetting (Rebuffi et al., 2017; Kang et al., 2022; Li & Hoiem, 2017; Hou et al., 2019; Buzzega et al., 2020; Dong et al., 2022; Douillard et al., 2020) or rectify the bias in the model (Wu et al., 2019; Belouadah & Popescu, 2019; Zhou et al., 2022a; Zhao et al., 2020). Unlike existing continual methods that do not consider data geometry and use Euclidean space for all data, we advocate using hyperbolic embeddings for continual learning to capture class-class and class-instance hierarchies.

While class-level continual learning has received significant attention, the problem of joint instance- and class-level continual learning remains relatively unexplored. The recent EgoObjects benchmark (Zhu et al., 2023), based on earlier benchmarks (Lomanco & Maltoni, 2017; She et al., 2020), makes it possible to investigate continual learning at the instance level. Incremental learning of instances has many applications, such as in (i) robotics, *i.e.,* recognizing and interacting with specific objects in the environment; (ii) in NLP, *i.e.,* identifying and classifying new instances of entities (*e.g.,* people, locations, organizations); (iii) in medical imaging, *i.e.,* identifying new forms of lesions and tumors; and (iv) in product recognition, *i.e.,* product recommendations, inventory management, and visual search. Parshotam & Kilickaya (2020) realize the need to incrementally recognize different instances of the same object class

using a metric learning approach under an object re-identification setting. In contrast, we propose continually learning both instance-level and class-level representations. According to the challenge winners (Pellegrini, 2022), iCaRL (Rebuffi et al., 2017) is the best-performing model for instance-level classification on the EgoObjects dataset. Therefore, our proposed hyperbolic continual learner takes inspiration from the core design choices of iCaRL.

### 2.2 Hyperbolic Learning

Hyperbolic learning has gained considerable attention in deep learning in embedding taxonomies and tree-like structures (Nickel & Kiela, 2017; Ganea et al., 2018a; Law et al., 2019; Nickel & Kiela, 2018), graphs (Liu et al., 2019; Chami et al., 2019; Bachmann et al., 2020; Dai et al., 2021), and text (Tifrea et al., 2018; Zhu et al., 2020; Dhingra et al., 2018; Leimeister & Wilson, 2018). Hyperbolic space is a space with constant negative curvature that can be thought of as a continuous version of a tree, making it a good choice to embed any finite tree while preserving the distances (Ungar, 2008; Hamann, 2018). Based on the tree-like behavior, Nickel & Kiela (2017) introduce a new approach to embed symbolic data in the Poincaré ball model, a particular hyperbolic space model. Ganea et al. (2018a) take a step forward and improve the Poincaré embeddings (Nickel & Kiela, 2017) using a model based on the geodesically convex entailment cones, showing the effectiveness when embedding data with a hierarchical structure. Furthermore, hyperbolic space has also been used to develop intermediate layers (Ganea et al., 2018a; Cho et al., 2019), and deep neural networks (Ganea et al., 2018b; Shimizu et al., 2020).

Following the initial success, hyperbolic space has shown success and gained attention in computer vision tasks in supervised and unsupervised learning (Mettes et al., 2024). Hyperbolic embeddings have shown benefits in classification and few-shot learning (Khrulkov et al., 2020; Gao et al., 2021; Guo et al., 2022b; Ghadimi Atigh et al., 2021b), zero-shot learning (Liu et al., 2020; Hong et al., 2023), segmentation (Atigh et al., 2022; Chen et al., 2023), out-of-distribution generalization (Ganea et al., 2018b; van Spengler et al., 2023), uncertainty quantification (Atigh et al., 2022; Chen et al., 2023), contrastive learning (Yue et al., 2023; Ge et al., 2023), hierarchical representation learning (Long et al., 2020; Dhall et al., 2020; Doorenbos et al., 2024), generative learning (Cho et al., 2024; Dai et al., 2020; Mathieu & Nickel, 2020), and vision-language representation learning (Ibrahimi et al., 2024; Desai et al., 2023). To our knowledge, Gao et al. (2023) is the only study that utilizes non-Euclidean geometries for class-level continual learning. Their approach introduces an expanding geometry of mixed-curvature space, particularly targeting low-memory scenarios. In contrast, we propose hyperbolic continual learning tailored for joint learning of instances and classes in a default memory regime.

## 3 HyperCLIC

In this section, we present our method for continual learning of instances and classes using hyperbolic geometry. Our goal is to continually learn hierarchy-aware representations, enabling classification at different levels of granularity: instance-, class-, and superclass-level. We follow a class-incremental learning setup, where new instance classes are introduced at each new task. Consider a sequence of $T$ training tasks $\mathcal{D}^1, \mathcal{D}^2, ..., \mathcal{D}^T$ with non-overlapping instances, where $\mathcal{D}^t = (x_i^t, y_i^t)_{i=1}^{n_t}$ is the $t$-th incremental step with $n_t$ training samples. Each $x_i^t \in \mathbb{R}^{\mathcal{D}}$ is an example of instance $y_i \in Y_t$, where $Y_t$ is the label space of task $t$. We formalize the hierarchy of instances, classes, superclasses, and other nodes as a tree $\mathcal{T} = (V, E)$. Each label in $Y_t$ corresponds to a leaf node label of tree $\mathcal{T}$, as detailed in Section 3.1. We only have access to samples of instances in $\mathcal{D}^t$ when training task $t$. Figure 2 illustrates the two main components of HyperCLIC: First, we embed the class-instance hierarchy into hyperbolic space using Poincaré embeddings and entailment cones, leveraging hyperbolic space's ability to model hierarchical relationships. Second, we perform continual hyperbolic alignment between visual inputs and the embedded hierarchy. We apply a hyperbolic prototype-based loss for classifying new instances and a hyperbolic distillation loss to maintain the consistency of embeddings for previously seen instances, ensuring that the model respects the hierarchical structure during continual learning. Our method consists of two stages. The objective of the first stage (Section 3.1) is to obtain a set of hyperbolic prototypes, which are later used in the second stage (Section 3.2) for the classification loss.

### 3.1 Embedding class-instance hierarchies in hyperbolic space

The joint class-instance tree hierarchy $\mathcal{T} = (V, E)$ (Figure 1) consists of four types of vertices $V = V_I \cup V_C \cup V_S \cup V_O$ and four types of directed edges $E = E_{IC} \cup E_{CS} \cup E_{SO} \cup E_{OO}$. Vertices $V_I =$

Figure 2: **Overview of HyperCLIC .** The class-instance hierarchy is projected into a fixed shared hyperbolic space. When learning an instance (e.g., instance A), samples of A go through the feature extractor and are mapped into the shared hyperbolic space using the exponential map. These samples are then pushed toward their hyperbolic instance prototype via classification loss and are encouraged to maintain hyperbolic logits from previous classes through distillation loss.

$\{i_1, i_2, ..., i_{n_I}\}$ are the set of instances, $V_C = \{c_1, c_2, ..., c_{n_C}\}$ the set of classes that are parents of instances, $V_S = \{s_1, s_2, ..., s_{n_S}\}$ the set of superclasses that are parents of classes, and $V_O = \{o_1, o_2, ..., o_{n_O}\}$ are the set of other remaining nodes. Each node $v \in V$ corresponds to a distinct label. The number of nodes $|V|$ equals the number of all labels $|C|$ in the hierarchy. $E_{IC} = \{(i_j, c_k) \mid i_j \in V_I, c_k \in V_C\}$ represent edges between instances and classes, $E_{CS} = \{(c_k, s_m) \mid c_k \in V_C, s_m \in V_S\}$ edges between classes and superclasses, $E_{SO} = \{(s_m, o_p) \mid s_m \in V_S, o_p \in V_O\}$ edges between superclasses and other nodes, and $E_{OO} = \{(o_p, o_q) \mid o_p \in V_O, o_q \in V_O\}$ represents the remaining 1-hop hypernymy relations. This formalization defines the hierarchical relationships in a tree structure where each node $V$ (instance, class, superclass, and other) except the root has exactly one parent. Each instance $V_I$ is a leaf node in the tree structure $\mathcal{T}$.

Given the class-instance tree hierarchy $\mathcal{T}$, our goal is to embed the symbolic tree representation *a priori* into a hyperbolic embedding space that incorporates the hierarchical relations between classes and from classes to instances. We define the hyperbolic manifold using the Poincaré ball model (Nickel & Kiela, 2017), as it is well-suited for gradient-based optimization. More formally, let $\mathbb{B}_c^d = \{z \in \mathbb{R}^d | c\|z\|^2 < 1\}$ be the open d-dimensional unit ball, where $\|.\|$ denotes the Euclidean norm, and $c$ denotes the curvature. The Poincaré ball model corresponds to the Riemannian manifold $(\mathbb{B}_c^d, g_z^{\mathbb{B}})$ with the Riemannian metric tensor $g_z^{\mathbb{B}} = 4(1 - c\|z\|^2)^{-2}\mathbf{I}_d$.

Let $d_{\mathcal{T}}(v_i, v_j)$ denote the graph distance between two nodes $v_i, v_j \in V$ based on their hierarchical relations. Also, let $d_{\mathbb{B}}(p_i, p_j)$ denote the hyperbolic distance between two points $p_i, p_j \in \mathbb{B}^d$ in the Poincaré ball. We seek to obtain a set of prototypes $P = \{p_i\}_{i=1}^{|V|}$ corresponding to $V = \{v_i\}_{i=1}^{|V|}$ such that $d_{\mathbb{B}}(p_i, p_j) \propto d_{\mathcal{T}}(v_i, v_j)$, *i.e.*, the hyperbolic distance $d_{\mathbb{B}}$ is proportional to the edge distances in graph $\mathcal{T}$. The hyperbolic distance $d_{\mathbb{B}}$ between two points $p_1, p_2 \in \mathbb{B}^d$ is given by the following equation:

$$d_{\mathbb{B}}(p_1, p_2) = \frac{2}{\sqrt{c}} \operatorname{arctanh}(\sqrt{c}\| - p_1 \oplus_c p_2\|). \tag{1}$$

where $\langle p_1, p_2 \rangle$ denotes the inner product of two vectors $p_1$ and $p_2$ and $\oplus$ denotes the Möbius addition given by:

$$p_1 \oplus_c p_2 = \frac{(1 + 2c\langle p_1, p_2 \rangle + c\|p_2\|^2)p_1 + (1 - c\|p_1\|^2)p_2}{1 + 2c\langle p_1, p_2 \rangle + c^2\|p_1\|^2\|p_2\|^2} \tag{2}$$

We minimize the loss function $\mathcal{L}_{Poincaré}$ defined by Nickel & Kiela (2017). Let $\mathcal{R} = \{(u, v)\}$ denote the set of transitive closures in graph $\mathcal{T}$, meaning there is a path from node $u$ to $v$. Ganea et al. (2018a) define $\mathcal{R}$ as entailment relations, where $v$ entails $u$, or equivalently, that $u$ is a subconcept of $v$. $\mathcal{L}_{Poincaré}$ encourages semantically similar objects to be close in the embedding space according to their Poincaré distance:

$$\mathcal{L}_{Poincaré} = \sum_{(u,v) \in \mathcal{R}} \log \frac{e^{-d_B(u,v)}}{\sum_{v' \in \mathcal{N}(u)} e^{-d_B(u,v')}}. \tag{3}$$

$\mathcal{N}(u) = \{v|(u,v) \notin \mathcal{R}\} \cup \{u\}$ denotes the set of negative examples for $u$. Negative examples $\mathcal{N}(u)$ include all nodes $v$ that do not entail $u$. While the Poincaré loss results in hyperbolic prototypes $P$ in tree-shaped regions on $\mathbb{B}^d$, there are no guarantees of entailment, *i.e.,* of a partial order relationship that requires the region of each subtree to be fully covered by their parent tree. Therefore, following Ganea et al. (2018a), we apply a max-margin entailment loss $\mathcal{L}_{Entailment}$ on the extracted Poincaré embeddings $P$, to enforce entailment regions:

$$\mathcal{L}_{Entailment} = \sum_{(u,v)\in\mathcal{R}} E(u,v) + \sum_{(u',v')\in\mathcal{N}} max(0, \gamma - E(u',v')). \tag{4}$$

The energy function $E(u,v) := \max(0, \Xi(u,v) - \psi(v))$ measures how far point $u$ from belonging to the entailment cone $\psi(v)$ is. The first term encourages $u$ to be part of the entailment cone $\psi(v)$ for $(u,v) \in \mathcal{R}$. The second term pushes negative samples $(u',v') \in \mathcal{N}$ angularly away for a minimum margin $\gamma > 0$ if they don't share an entailment cone. For full details of Equation 4, we refer to Ganea et al. (2018a).

We observe that the entailment loss tends to bring the prototypes, especially those of the instances $V_I$, too close to each other. To counteract this, we apply a separation loss $\mathcal{L}_S$, similar to the approach in Long et al. (2020), which ensures that all prototypes $P$ are adequately separated.

$$\mathcal{L}_S(P) = \vec{1}^T(\bar{P}\bar{P}^T - I)\vec{1}, \tag{5}$$

where $\bar{P}$ denotes the vector-wise $l_2$-normalization of $P$. The proposed loss function minimizes the cosine similarity between any two prototypes. Algorithm 1 summarizes the first stage of HyperCLIC, detailing the sequential application of the three losses that result in the hyperbolic prototypes. After obtaining the prototypes for all nodes in the tree $\mathcal{T}$, we only use the prototypes $P = \{P_y\}_{y=1}^{|V_I|}$ corresponding to the instance (leaf) nodes $V = \{v_y\}_{y=1}^{|V_I|}$ in the next section.

---

**Algorithm 1**
3.1 Embedding class-instance hierarchies in hyperbolic space

---

**Require:** The joint class-instance hierarchy $\mathcal{T} = (V, E)$

1: We seek to obtain a set of prototypes $P = \{p_i\}_{i=1}^{|V|}$ corresponding to $V = \{v_i\}_{i=1}^{|V|}$ such that $d_{\mathbb{B}}(p_i, p_j) \propto d_{\mathcal{T}}(v_i, v_j)$
2: // Initialize prototypes with Poincaré loss:
3: **for** $e$ in `#Poincaré_Epochs` **do**
4:    Minimize loss: $\mathcal{L}_{Poincaré} = \sum \log \frac{e^{-d_B(u,v)}}{\sum_{v' \in \mathcal{N}(u)} e^{-d_B(u,v')}}$
5: **end for**
6: // Enforce max-margin entailment regions:
7: **for** $e'$ in `#Entailment_Epochs` **do**
8:    Minimize loss: $\mathcal{L}_{Entailment} = \sum E(u,v) + \max(0, \gamma - E(u',v'))$
9: **end for**
10: // Enforce separation between prototypes:
11: **for** $e''$ in `#Separation_Epochs` **do**
12:    Minimize loss: $\mathcal{L}_S(P) = \vec{1}^T(\bar{P}\bar{P}^T - I)\vec{1}$
13: **end for**
14: After obtaining the prototypes for all nodes in the tree $\mathcal{T}$, we only use the prototypes $P = \{P_y\}_{y=1}^{|V_I|}$ corresponding to the instance (leaf) nodes $V = \{v_y\}_{y=1}^{|V_I|}$ in the next stage.

---

---

**Algorithm 2**
3.2 Continual hyperbolic learning with hierarchical prototypes

---

**Require:** $P = \{P_y\}_{y=1}^{|V_I|}$

1: // The classification loss using prototypes from the previous stage:

2: $\mathcal{L}_{cls} = -\frac{1}{|\mathcal{D}^t|} \sum_{(x_i, y_i) \in \mathcal{D}^t} \sum_{i=1}^{|C|} y_i \cdot \log\left( \frac{e^{h(z_i^t, y_i)}}{\sum_{j=1}^{|C|} e^{h(z_i^t, y_j)}} \right)$

3: Keep a copy of the model parameters at time $t-1$ as $\theta^{t-1}$ to calculate the distillation loss:

4: $\mathcal{L}_{distil} = -\frac{1}{|\mathcal{D}^{b<t}|} \sum_{(x_i, y_i) \in \mathcal{D}^{b<t}} \sum_{i=1}^{|C^{b<t}|} p(y_i | z_i^{t-1}) \cdot \log p(y_i | z_i^t)$

5: Run network training by minimizing $\mathcal{L} = \mathcal{L}_{cls} + \mathcal{L}_{distil}$

---

### 3.2 Continual hyperbolic learning

We use a backbone $\varphi(.; \theta_t)$ to extract features from a training sample $x_i$, where $\theta_t$ denotes the model parameters at timestep $t$. The extracted features form a Euclidean representation $\varphi(x_i; \theta_t)$. We assume that the backbone output is in the tangent space $T_x\mathcal{M}$, while the extracted prototypes from Section 3.1 are in the hyperbolic space $\mathcal{M}$. Thus, we need to project the Euclidean representation to the hyperbolic space $\mathcal{M}$ through the exponential map function.

$$z_i^t = \exp_0\left( \varphi(x_i; \theta^t) \right). \tag{6}$$

Here, $z_i^t$ is the hyperbolic representation of $\varphi(x_i; \theta_t)$. The exponential map function is given as (Ganea et al., 2018b):

$$\exp_0^c(x) = \tanh\left( \sqrt{c}\|x\| \right) \cdot \frac{x}{\sqrt{c}\|x\|}, \tag{7}$$

where $c$ denotes the curvature of the hyperbolic space. A higher $c$ indicates a more curved space. The exponential map embeds visual representation into the hyperbolic space where the hierarchy of classes and instances are embedded as prototypes. Our goal is to minimize the hyperbolic distance of representation z with its instance prototype $P_y$. Thus, we define the hyperbolic logit $h(z, y)$ as the negative hyperbolic distance of vector representation z and all prototypes $y$:

$$h(z, y) = -\frac{d_{\mathbb{B}}(z, P_y)}{\tau}. \tag{8}$$

The temperature parameter $\tau$ controls the entropy of the probability distribution while preserving the relative ranks of each class. We can make the probability distribution more peaked or smoother by adjusting this parameter. The conditional probability $p(y_i | z_i^t)$ can be derived by the softmax of the hyperbolic logit:

$$p(y_i | z_i^t) = \frac{e^{h(z_i^t, y_i)}}{\sum_{j=1}^{|C|} e^{h(z_i^t, y_j)}}. \tag{9}$$

Once the hyperbolic logits are calculated, we can compute the classification loss as the cross entropy between the hyperbolic logits and the targets:

$$\mathcal{L}_{cls} = -\frac{1}{|\mathcal{D}^t|} \sum_{(x_i, y_i) \in \mathcal{D}^t} \sum_{i=1}^{|C|} y_i \cdot \log\left( \frac{e^{h(z_i^t, y_i)}}{\sum_{j=1}^{|C|} e^{h(z_i^t, y_j)}} \right). \tag{10}$$

While the classification loss ensures that the new classes learn their representation, the ultimate goal of class-incremental learning is to continually learn a model that works both for the old and the new classes. Formally, the model should not only acquire the knowledge from the current task $\mathcal{D}^t$ but also preserve the knowledge from former tasks $\mathcal{D}^{b<t}$. We use a hyperbolic distillation loss to ensure that the model predicts the same hyperbolic logits for the old classes $\mathcal{D}^{b<t}$ at time $t$ as time $t-1$. Following Rebuffi et al. (2017), we keep a copy of the model parameters at time $t-1$ as $\theta^{t-1}$ and define the probability distribution at time $t-1$ as $p(y_i | z_i^{t-1})$ where $(x_i, y_i) \in \mathcal{D}^{b<t}$:

$$p(y_i | z_i^{t-1}) = \frac{e^{h(z_i^{t-1}, y_i)}}{\sum_{j=1}^{|C|} e^{h(z_i^{t-1}, y_j)}}. \tag{11}$$

The hyperbolic distillation loss is calculated as the cross-entropy loss between the probability distribution at time $t$ and $t-1$:

$$\mathcal{L}_{distil} = -\frac{1}{|\mathcal{D}^{b<t}|} \sum_{(x_i,y_i)\in\mathcal{D}^{b<t}} \sum_{i=1}^{|C^{b<t}|} \left(\frac{e^{h(z_i^{t-1},y_i)}}{\sum_{j=1}^{|C^{b<t}|} e^{h(z_i^{t-1},y_j)}}\right) \cdot \log\left(\frac{e^{h(z_i^t,y_i)}}{\sum_{j=1}^{|C^{b<t}|} e^{h(z_i^t,y_j)}}\right). \tag{12}$$

The final loss of the second stage is $\mathcal{L} = \mathcal{L}_{cls} + \mathcal{L}_{distil}$ following Rebuffi et al. (2017). Algorithm 2 summarizes the second stage of HyperCLIC, where the hyperbolic prototypes from the first stage are used to classify new classes at the current task and distill hyperbolic logits from previously seen classes. Samples from old classes $\mathcal{D}^{b<t}$ are selected at the end of each task by the herding strategy (Welling, 2009), which is a commonly used strategy aiming to select the most representative samples of each class. These exemplars are saved in memory and are representative data points of each known instance class. In addition to distillation loss, the exemplars are also used during inference. During inference, the closeness to exemplars determines the final prediction for a given data point. We perform the nearest mean of exemplar classification following Rebuffi et al. (2017):

$$y_i^* = \underset{y=1,\dots,n_I}{\arg\min} \|\varphi(\mathrm{x_i}) - \mu_y\| \tag{13}$$

Where $\mu_y$ denotes the mean of exemplars for the instance class $y$ and $n_I$ denotes the number of instance classes. The exemplars and their means could be calculated in hyperbolic space, however, we observe that the Euclidean representations are already aligned with the hierarchy and conclude that there is no need for an extra hyperbolic computation. Overall, our method aligns classification and distillation losses with hyperbolic representations of a fixed hierarchy, leveraging the class-instance hierarchy with minor modifications to existing continual learning methods. This alignment allows continual classification at multiple levels of granularity such as instance and class-level.

## 4 Experimental Setup

**Datasets** We conduct experiments using the class-incremental instance-level continual learning benchmark EgoObjects (Zhu et al., 2023), first introduced at the 3rd CLVISION Challenge (Pellegrini et al., 2022). This dataset comprises a stream of 15 tasks, each created by cropping the main object from egocentric videos. In total, there are 1110 unique instances from 277 classes. Since the test set labels are not provided in the challenge, we split the training set into training (80%), testing (10%), and validation (10%) subsets. Following the challenge convention, we divide the dataset into non-overlapping sets of `gaia_ids`, which are identifiers for each unique video clip. We also conduct experiments on CORe50 NC scenario (Lomonaco et al., 2020; Lomanco & Maltoni, 2017) that is specifically designed for continual object recognition with 50 domestic object instances belonging to 10 categories under different backgrounds and lighting. Additionally, we experiment with iCIFAR-100 (Krizhevsky et al., 2009; Rebuffi et al., 2017), which contains $i = 10$ tasks, each with 10 classes. Although CIFAR-100 is designed for class-level classification, has limited hierarchy and lacks instances within classes, we treat the classes as instances to facilitate comparison with other class-incremental continual learning methods.

Our focus is real-world instance-level continual learning. For this research problem, EgoObjects is the only large-scale dataset. The Core50 dataset is also an instance-level dataset, but is a toy dataset with only 50 instances, resulting in minimal differences between methods. We have added Core50 with a comparison to iCaRL (the closest Euclidean counterpart to our method) to Table 9. We have moved the CIFAR-100 8 and Core50 9 experiments to the Appendix (7) to keep the focus on EgoObjects as the only relevant benchmark. We see a great potential for additional large instance-level continual datasets due to the real-world relevance, but there is unfortunately a lack of large-scale instance-level continual datasets.

**Hierarchies** For the EgoObjects (Zhu et al., 2023) dataset, we construct a hierarchy using Word-Net (Fellbaum, 2010), a comprehensive lexical database of the English language. We match each class in the dataset with the most frequent noun synset in WordNet. To find the path from the leaf nodes to the root node, we iteratively choose the hypernym with the shortest path to the root. If this path does not include the `physical_entity.n.01` node, we select the next shortest path that does. We manually add classes that do not match any synsets to the hierarchy. Categories without any instances (91 out of 277)

are removed. We also remove nodes with only one child, except for parents of leaf nodes. The resulting unbalanced hierarchy has a maximum of 7 instances per category, a minimum of 1, and a depth of 12. Our automated hierarchy extraction method can be adapted to other datasets. In CIFAR-100 (Krizhevsky et al., 2009), classes are grouped into 20 superclasses. Each image has a *fine* label (the class) and a *coarse* label (the superclass). Following Yan (2021), we group the 20 superclasses into 7 ultra classes that are children of the root node. In CORe50, the dataset consists of 10 classes, each containing 5 instances. To extend the hierarchy, we group the 10 classes into 6 superclasses, which are further grouped into 3 ultra classes. These 3 ultra classes are then made children of a single root node.

**Implementation details** For HyperCLIC EgoObjects experiments, we train a ResNet34 backbone for 3 epochs following Pellegrini (2022), with a batch_size of 90, and a learning rate of 0.01 for 15 tasks. The memory size is 3500 according to the challenge. The pretrained models are pretrained with `ResNet34_Weights.IMAGENET1K_V1`, `Wide_ResNet50_2_Weights.IMAGENET1K_V2` and `RegNet_X_16GF_Weights.IMAGENET1K_V2`. In all datasets, the curvature of the Poincaré model (`PoincaréBallExact` Kochurov et al. (2020)) is 1, and the temperature $\tau$ is 10. The hyperbolic prototypes are 64-dimensional using Ganea et al. (2018a) with 150 Poincaré epochs and 50 entailment epochs, plus separation following Long et al. (2020) with a learning rate of 1.0 for 500 epochs. In the iCIFAR-100 experiments, we train a ResNet34 backbone for 100 epochs with a batch size of 128, starting with a learning rate of 2.0 for iCaRL and 0.01 for HyperCLIC. A MultiStepLR scheduler with a gamma of 0.2 is applied after every 50 epochs. The memory size is set to 2000 for the 100 classes. The hyperbolic prototypes are created in the same way as in EgoObjects. For the pretrained experiments, our method and baseline are trained for 6 epochs. In the CORe50 experiments, we train a ResNet34 backbone for 3 epochs with a batch size of 90. The learning rate is 0.01 for HyperCLIC and 2.0 for iCaRL. The memory size is 1000, with 20 samples per instance, as in the iCIFAR-100 experiments. The hyperbolic prototypes are 10-dimensional and are created using the method in Ganea et al. (2018a), with 150 Poincaré epochs and 50 entailment epochs.

**Evaluation metrics** We perform the evaluation on both standard and continual hierarchical metrics. The standard metrics include average forgetting, average accuracy, and per-task accuracies. In our work, the accuracy metric reflects only the instance-level accuracy, so we refer to it as the continual instance-level accuracy. The instance-level accuracy on task $t$ after incremental learning of task $t'$ is defined as:

$$\texttt{Acc}_{instance}^{tt'} = \frac{1}{n_t} \sum_{i=1}^{n_t} 1\{\hat{y_i}^{tt'} = y_i^t\} \tag{14}$$

where $\hat{y_i}^{tt'}$ is the predicted class for instance label $y_i^t$ evaluated on the test set of $t$-th task after incremental learning of the $t'$-th task. To evaluate continual hierarchical consistency and robustness, we report continual class-level and superclass-level accuracies, inspired by sibling and cousin accuracy in Ghadimi Atigh et al. (2021a). For each sample $x_i^t$ and its ground-truth instance label $y_i^t$, let $p(y_i^t)$ be the parent class $y_i^t$ and $gp(y_i^t)$ be the grandparent class of $y_i^t$. In class-level accuracy, a prediction is also correct if it shares a parent with the target class. The class-level accuracy on task $t$ after incremental learning of task $t'$ is defined as:

$$\texttt{Acc}_{class}^{tt'} = \frac{1}{n_t} \sum_{i=1}^{n_t} 1\{p(\hat{y_i}^{tt'}) = p(y_i^t)\} \tag{15}$$

In the superclass-level accuracy, the predicted labels must share a grandparent with the target class to count as correct. The superclass-level accuracy on task $t$ after incremental learning of task $t'$ is defined as:

$$\texttt{Acc}_{superclass}^{tt'} = \frac{1}{n_t} \sum_{i=1}^{n_t} 1\{gp(\hat{y_i}^{tt'}) = gp(y_i^t)\} \tag{16}$$

Similar to Bertinetto et al. (2020); Garg et al. (2022a), we also report the continual distance to the Lowest Common Ancestor (LCA) that captures the mistake's hierarchical severity. For the wrong predictions, LCA distance reports the average number of edges between the predicted node and the LCA of the predicted and ground-truth nodes. LCA distance reveals how hierarchically far off the inaccurate predictions are, considering the joint class-instance hierarchy as the ground truth.

Table 1: **Experimental results on EgoObjects** with a ResNet34 backbone. Both with and without pre-training, HyperCLIC performs best.

| | Pretrain | Accuracy ↑ | | | LCA ↓ | Forgetting ↓ |
| | | Instance | Class | Superclass | | |
|---|---|---|---|---|---|---|
| Naive (fine-tune) | | 0.44 | 1.07 | 1.58 | 5.74 | 10.38 |
| EWC (Kirkpatrick et al., 2017) | | 0.22 | 0.66 | 1.42 | 5.26 | 10.38 |
| iCaRL (Rebuffi et al., 2017) | | 20.05 | 21.39 | 22.24 | 5.45 | 10.57 |
| CoPE (De Lange & Tuytelaars, 2021) | | 3.14 | 4.45 | 4.91 | 5.63 | 25.38 |
| GDumb (Prabhu et al., 2020) | | 0.50 | 1.27 | 1.88 | 5.50 | 5.87 |
| DER (Yan et al., 2021) | | 19.59 | 20.59 | 21.14 | 5.61 | 35.39 |
| HyperCLIC | | **41.76** | **45.91** | **48.04** | **4.93** | **4.17** |
| Naive (fine-tune) | ✓ | 6.06 | 15.55 | 18.24 | 4.70 | 93.03 |
| EWC (Kirkpatrick et al., 2017) | ✓ | 6.07 | 15.77 | 18.37 | 4.74 | 92.83 |
| iCaRL (Rebuffi et al., 2017) | ✓ | 81.63 | 87.02 | 87.81 | 3.89 | **3.77** |
| CoPE (De Lange & Tuytelaars, 2021) | ✓ | 37.15 | 49.89 | 52.51 | 4.22 | 32.40 |
| GDumb (Prabhu et al., 2020) | ✓ | 61.51 | 71.00 | 72.82 | 4.02 | 18.22 |
| DER (Yan et al., 2021) | ✓ | 80.00 | 84.40 | 85.20 | 4.28 | 10.59 |
| HyperCLIC | ✓ | **84.81** | **91.94** | **92.67** | **2.99** | 7.08 |

Following the 3rd CLVision challenge (Verwimp et al., 2023) and the SSLAD competition (Pellegrini et al., 2022)), all continual hierarchical metrics are calculated as the average mean over all the tasks. Here, we take the continual instance-level accuracy as an example. Let $Acc_{instance}^{ij} \in [0, 1]$ denote the instance-level classification accuracy evaluated on the test set of the $i$-th task after incremental learning of the $j$-th task. The average mean instance-level accuracy is defined as:

$$\texttt{Accuracy}_{\texttt{instance-level}} = \frac{1}{T^2} \sum_{j=1}^{T} \sum_{i=1}^{T} Acc_{instance}^{ij} \tag{17}$$

where $T$ denotes the number of all tasks. We use the same formulation for average mean class-level accuracy and superclass-level accuracies. We also report the per-task accuracies at the end of the last task.

## 5 Results

### 5.1 Comparison to existing methods

**Hierarchical and continual metrics** Table 1 reports the results of different baselines on the EgoObjects dataset. If trained from scratch, HyperCLIC achieves a significantly higher instance-level accuracy (41.76%), class-level accuracy (45.91%), and superclass-level accuracy (48.04%) compared to all baselines. The closest competitor is iCaRL (Rebuffi et al., 2017) with an instance-, class-, and superclass-level accuracy of 20.05%, 21.39%, and 22.24%, respectively, which is less than half of HyperCLIC 's performance. This demonstrates HyperCLIC's strength in maintaining performance across different levels of granularity. HyperCLIC shows a lower LCA value (4.93) indicating that the inaccurate predictions are less hierarchically severe compared to other baselines. The second-best is EWC (Kirkpatrick et al., 2017) with 5.26. HyperCLIC also shows the lowest forgetting rate (4.17%), indicating better retention of learned knowledge. For the pretrained scenario, HyperCLIC again leads with the instance-level accuracy of 84.81%, followed by iCaRL (Rebuffi et al., 2017) with 81.63%, and DER(Yan et al., 2021) with 80%. DER is a strong non-transformer-based continual baseline. In both trained-from-scratch and pretrained scenarios, HyperCLIC consistently outperforms all baseline methods across all metrics. It excels in instance-, class-, and superclass-level accuracies while showing lower LCA values and minimal forgetting. This indicates that HyperCLIC not only learns a hierarchically-aware representation but also retains knowledge from previous tasks, proving its superiority and robustness over other existing methods.

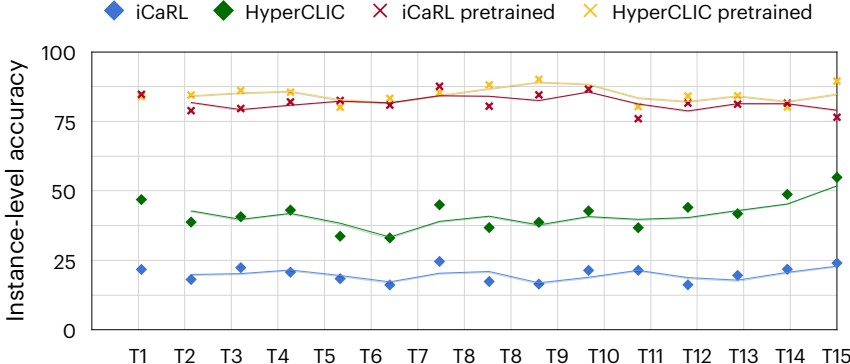

Figure 3: **The final per-task results and their moving average on EgoObjects** with a ResNet34 backbone. HyperCLIC outperforms iCaRL by maintaining prior knowledge while delivering higher accuracy on new tasks.

Given that iCaRL (Rebuffi et al., 2017) has the highest performance after HyperCLIC, we use it as the primary comparison for evaluating our method in subsequent experiments. Figure 3 illustrates the instance-level accuracy for each task (T1 to T15) after the completion of all tasks. In both non-pretrained and pretrained models, iCaRL (Rebuffi et al., 2017) demonstrates higher stability but at the cost of not learning effective representations for new tasks, thereby compromising its performance on recent tasks to retain knowledge from previous ones. We can also observe this behavior in Table 1. In contrast, our method not only preserves knowledge from prior tasks but also achieves higher accuracy on the latest tasks compared to the baseline.

## 5.2 Ablation study

**Quality of the hyperbolic prototypes**   In the first stage of HyperCLIC, we focus on building high-quality prototypes that could later be used in the second stage of our method. Thus, we conduct an ablation on the most critical hyperparameters that define the quality of the prototypes: the hyperbolic embedding dimension and the number of entailment epochs. The results, presented in Table 2, show that a very small embedding dimension fails to capture an effective hyperbolic representation. However, performance improves with medium-sized and larger dimensions, which are competitive. Additionally, we observed that increasing the number of entailment epochs negatively impacts performance. This occurs because the entailment loss pushes the children of the same parent too close to each other, leading to a deterioration in both instance-level accuracy and hierarchical metrics.

Table 2: **The effect of different embedding dimensions and entailment epochs on HyperCLIC in CORe50.** Very small dimension sizes cannot capture the hierarchical representation. For larger dimensions, the number of entailment epochs becomes a significant factor: too few epochs cannot enforce the entailment cone, while too many epochs collapse leaf nodes into each other.

| Dimension | Entailment epochs | Instance | Class | Superclass | LCA | Forgetting |
|---|---|---|---|---|---|---|
| 64 | 150 | 20.48 | 35.56 | 45.58 | 2.98 | -10.67 |
|    | 50  | 21.13 | 37.94 | 48.58 | 2.91 | -9.17 |
| 10 | 150 | 21.05 | 34.53 | 46.84 | 2.99 | -12.19 |
|    | 50  | 22.89 | 36.80 | 47.13 | 3.01 | -11.59 |
| 5  | 150 | 10.48 | 22.51 | 30.59 | 3.21 | -2.82 |
|    | 50  | 12.05 | 22.58 | 32.24 | 3.26 | -5.10 |

**Temperature**   In the second stage of HyperCLIC, one of the key hyperparameters that significantly impacts the continual learning of new classes in the classification loss is the temperature. The temperature, $\tau$, controls the smoothness of the probability distribution derived from the pairwise distances of the hyperbolic prototypes. In hyperbolic literature (Ibrahimi et al., 2024; Long et al., 2020), $\tau$ is commonly

Table 3: **The effect of different temperatures on HyperCLIC in EgoObjects.** Our setting with temperature of 0.1 is highlighted in gray. Our method is more sensitive to the temperature in the from-scratch setting.

| | From scratch | | | | | Pretrained | | | | |
|------|----------|-------|------------|------|------------|----------|-------|------------|------|------------|
| | Instance | Class | Superclass | LCA | Forgetting | Instance | Class | Superclass | LCA | Forgetting |
| 0.01 | 26.14 | 28.26 | 29.18 | 5.38 | -1.81 | 77.79 | 84.90 | 85.64 | 3.92 | 8.06 |
| 0.05 | 11.79 | 12.93 | 13.78 | 5.54 | 4.52 | 82.21 | 90.02 | 91.16 | 2.97 | 6.67 |
| 0.09 | 38.21 | 42.59 | 44.44 | 4.83 | 3.54 | 82.03 | 89.93 | 91.13 | 2.91 | 5.21 |
| Ours | 41.76 | 45.91 | 48.04 | 4.93 | 4.17 | 84.81 | 91.94 | 92.67 | 2.99 | 7.08 |
| 0.3 | 19.76 | 24.07 | 27.22 | 4.94 | 11.82 | 79.22 | 88.76 | 89.88 | 2.88 | 7.70 |
| 0.5 | 16.35 | 20.77 | 23.68 | 4.97 | 9.11 | 76.80 | 87.35 | 88.72 | 2.74 | 6.34 |
| 1.0 | 12.67 | 16.52 | 19.21 | 5.03 | 10.39 | 71.86 | 85.35 | 87.21 | 2.62 | 6.47 |

Table 4: **EgoObjects results with different distillation losses** with a ResNet34 backbone. Cross-entropy loss consistently outperforms other distillation losses in both scenarios.

| | From scratch | | | | | Pretrained | | | | |
|---------------|----------|-------|------------|------|------------|----------|-------|------------|------|------------|
| | Instance | Class | Superclass | LCA | Forgetting | Instance | Class | Superclass | LCA | Forgetting |
| KL divergence | 32.60 | 35.30 | 36.95 | 5.13 | -1.40 | 49.76 | 62.45 | 65.46 | 3.75 | -6.00 |
| MSE | 36.09 | 40.33 | 42.53 | 4.94 | 8.18 | 84.38 | 91.21 | 92.33 | 2.92 | 7.84 |
| Cross-entropy | 41.76 | 45.91 | 48.04 | 4.93 | 4.17 | 84.11 | 91.61 | 92.54 | 2.96 | 7.08 |

set to 0.1. Our results, as shown in Table 3, indicate that HyperCLIC is highly sensitive to the choice of temperature, especially when trained from scratch compared to using a pretrained model. The best performance is consistently achieved with a temperature of 0.1 in both scenarios. When the probability distribution is either too peaked or too smooth, the model struggles to effectively distinguish between classes.

**Distillation loss** A key component of our method is the distillation loss, which ensures that the model retains hyperbolic knowledge from previous tasks. To evaluate this, we tested three different loss functions: Kullback-Leibner (KL) divergence, Mean Squared Error (MSE), and cross-entropy loss. Table 4 presents the results for both pretrained and from-scratch scenarios. KL divergence shows the lowest forgetting rates, at $-1.40\%$ and $-6\%$ respectively, for from-scratch and pretrained scenarios. However, it performs the worst in terms of hierarchical metrics and instance accuracy, suggesting it disrupts the balance between plasticity and stability by favoring stability at the expense of plasticity. Cross-entropy loss performs the best across both scenarios, which is why our method is based on this loss function. There is also a more notable difference between MSE and cross-entropy when training from scratch compared to the pretrained scenario.

**Balancing Classification and Distillation Losses** The continual learning objective in the second stage of HyperCLIC combines classification and distillation losses, following the approach introduced

Table 5: **Balancing the effect of classification and distillation losses in EgoObjects dataset.** Equally weighting the losses yields the highest performance in HyperCLIC.

| $\lambda$ | Accuracy ↑ | | | LCA ↓ | Forgetting ↓ |
|------|----------|-------|------------|------|-------|
| | Instance | Class | Superclass | | |
| 0.1 | 35.33 | 39.71 | 42.09 | 4.90 | 7.28 |
| 0.3 | 37.54 | 42.05 | 44.51 | 4.83 | 6.18 |
| Ours | 41.76 | 45.91 | 48.04 | 4.93 | 4.17 |
| 0.7 | 38.02 | 41.56 | 43.38 | 4.92 | -0.59 |
| 0.9 | 24.49 | 26.55 | 27.83 | 5.32 | -4.92 |

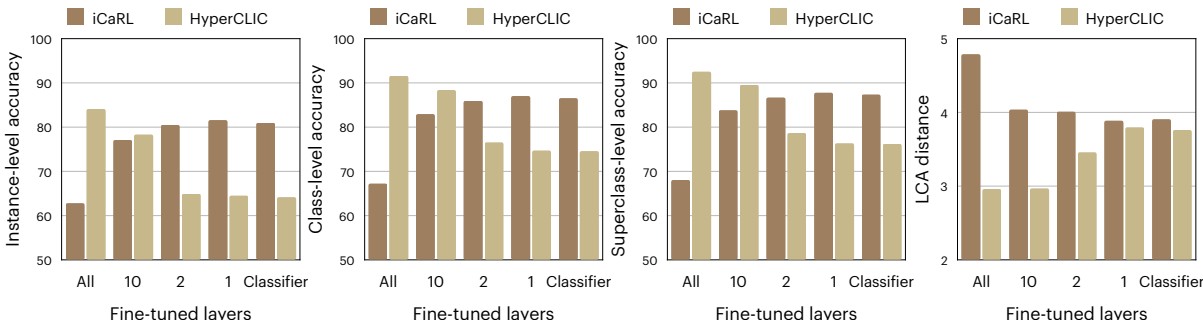

Figure 4: **Performance of HyperCLIC and iCaRL across four metrics** with varying numbers of fine-tuned layers. HyperCLIC excels when all layers are fine-tuned, while iCaRL performs better with more fixed layers. The Table 1 comparison in the pretrained scenario considers the optimal setup for each model.

Table 6: **Comparison of iCaRL and HyperCLIC on EgoObjects with various pretrained backbones.** HyperCLIC consistently outperforms iCaRL regardless of the architecture, as shown with ResNet34 (highlighted in gray). This demonstrates that HyperCLIC is adaptable to any backbone and, when paired with advanced models, enhances joint continual learning of instances and classes.

| | iCaRL | | | | | HyperCLIC | | | | |
|---|---|---|---|---|---|---|---|---|---|---|
| | Instance | Class | Superclass | LCA | Forgetting | Instance | Class | Superclass | LCA | Forgetting |
| WideResNet50 | 82.17 | 87.46 | 88.08 | 4.02 | 3.42 | 86.50 | 94.17 | 94.61 | 2.58 | 6.01 |
| RegNetX_16GF | 78.63 | 86.64 | 87.58 | 3.55 | 5.10 | 87.22 | 94.11 | 95.01 | 2.59 | 5.99 |
| ResNet34 | 81.63 | 87.02 | 87.81 | 3.89 | 3.77 | 84.81 | 91.94 | 92.67 | 2.99 | 7.08 |

in iCaRL. To evaluate the impact of each loss term, we use a balancing factor, $\lambda$, as shown in the loss function: $\mathcal{L} = \lambda \cdot \mathcal{L}_{distil} + (1 - \lambda) \cdot \mathcal{L}_{cls}$. By varying $\lambda$, we assess the influence of classification and distillation losses on the model's performance. Table 5 presents the results for different $\lambda$ values. Our findings indicate that a fully balanced loss achieves the best performance across both instance-level accuracy and hierarchical metrics. This highlights the importance of giving equal weight to both losses to ensure that the model can effectively learn new classes while retaining knowledge of previously learned classes.

**Number of fine-tuned layers**  We observed that the results of our method and iCaRL differ based on the number of layers fine-tuned in the pretrained scenario. Figure 4 illustrates the impact on four hierarchical metrics as the number of fine-tuned layers varies for both methods. For iCaRL, increasing the number of fine-tuned layers results in worse performance, with lower instance-, class-, and superclass-level accuracies, and higher LCA distance. Conversely, for HyperCLIC, more fine-tuned layers lead to improved results. We hypothesize that this is because HyperCLIC enforces a hierarchical structure on the representations, and since the pretrained model was trained in Euclidean space, fine-tuning more layers allows HyperCLIC to leverage the model's capacity to learn hierarchically-aware representations from the first layer. Given these trends, we use the best scores for each method to compare HyperCLIC with the baseline in the pretrained scenario.

**Backbones**  All results reported thus far utilize the ResNet34 backbone. Table 6 compares our method with the leading baseline, iCaRL (Rebuffi et al., 2017), using various pretrained backbones. The findings indicate that HyperCLIC consistently surpasses the baseline across the hierarchical metrics when using WideResNet50, RegNetX, and ResNet34 as the backbone architectures. Notably, HyperCLIC achieves the highest performance with the RegNet backbone, attaining instance-, class-, and superclass-level accuracies of 87.22%, 94.11%, and 95.01%, respectively, compared to the best iCaRL results for WideResNet, which are 82.17%, 87.46%, and 88.08%. Although HyperCLIC improves the LCA distance by nearly one edge, it shows a higher forgetting rate. These results suggest that our method, particularly when combined with advanced models such as transformers, has the potential to significantly enhance the joint continual learning of instances and classes.

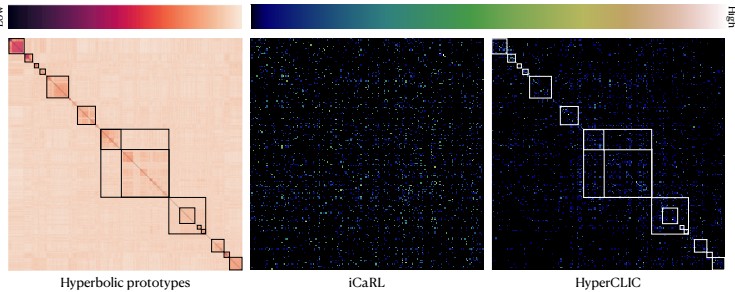

Figure 5: **Left:** the pair-wise distances of the instance-level hyperbolic prototypes. **Middle & right:** the class-level predictions only for the wrong instance-level predictions. The squares highlight the hierarchical structure of HyperCLIC mistakes compared with iCaRL and its similarity to the original hierarchy.

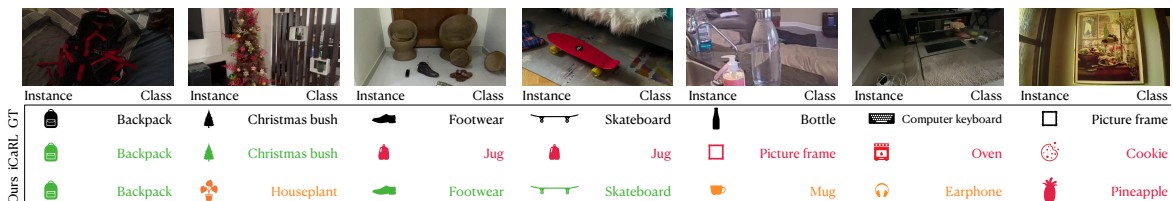

Figure 6: **Qualitative examples from EgoObjects** comparing instance and class-level predictions for ground truth ($1^{st}$ row), iCaRL ($2^{nd}$ row), and our method ($3^{rd}$ row). Green indicates correct predictions, while red and orange denote severe and less severe hierarchical errors, respectively. HyperCLIC makes less severe (orange) errors compared to iCaRL.

**Class-level errors** Figure 5 compares the class-level confusion matrix for incorrectly predicted instances between iCaRL and HyperCLIC, alongside the distances of hyperbolic prototypes. In this matrix, the squares represent children of the same parent node, and larger squares encompass multiple subtrees. The aim of this experiment is to demonstrate that even when instances are misclassified, HyperCLIC's class-level predictions still follow the hierarchical structure. As shown, iCaRL's class-level predictions are evenly distributed, while HyperCLIC's predictions are primarily concentrated along the diagonal. Additionally, a similar hierarchical organization is observed in HyperCLIC when compared with the pairwise hyperbolic prototype distances, represented by matching squares. This indicates that HyperCLIC maintains the hierarchical structure, even when making incorrect predictions, resulting in less severe errors.

**Qualitative examples: success and failure cases** Figure 6 presents several images from the test set and compares the instance and class-level predictions of our method against the baseline. In the far-left example, both methods easily make correct instance-level predictions. Whereas, the far-right example is challenging for both methods, as it depicts food items in a drawing, leading both to predict unrelated instances and classes. The middle examples reveal that when HyperCLIC makes an incorrect instance-level prediction, the class-level prediction remains hierarchically close. For example, it might mistake a "Christmas bush" for a "house plant" or a "bottle" for a "mug". This suggests that by incorporating hierarchical information during training, HyperCLIC reduces the severity of hierarchical errors.

**Statistical Significance** To evaluate the generality and stability of our method across different runs, we compare HyperCLIC with the best-performing baselines, iCaRL and DER, in both from-scratch and pretrained settings. Each experiment is conducted using 5 different random seeds, and we report the average and standard deviation of the results in Appendix Table 10. Additionally, we perform statistical significance testing to compare HyperCLIC with the baselines. Table 7 presents the p-values for each metric. The results demonstrate that there are statistically significant differences between HyperCLIC and the two baselines across all metrics. Specifically, all p-values are below the commonly used threshold of $p < 0.05$, confirming that the observed performance improvements are statistically significant.

Table 7: **Statistical Significance Testing (p-values) of HyperCLIC Compared with iCaRL and DER in From-Scratch and Pretrained Settings.** HyperCLIC demonstrates statistical significance compared to iCaRL and DER across all metrics, with a threshold of $p < 0.05$.

| | From scratch | | | | |
| --- | --- | --- | --- | --- | --- |
| | Instance | Class | Superclass | LCA | Forgetting |
| iCaRL | $3.80 \cdot 10^{-7}$ | $1.46 \cdot 10^{-7}$ | $1.45 \cdot 10^{-7}$ | $1.19 \cdot 10^{-9}$ | $1.59 \cdot 10^{-4}$ |
| DER | $1.53 \cdot 10^{-6}$ | $5.79 \cdot 10^{-7}$ | $3.96 \cdot 10^{-7}$ | $2.71 \cdot 10^{-6}$ | $2.84 \cdot 10^{-10}$ |
| | Pretrained | | | | |
| | Instance | Class | Superclass | LCA | Forgetting |
| iCaRL | $2.84 \cdot 10^{-2}$ | $1.61 \cdot 10^{-5}$ | $4.98 \cdot 10^{-6}$ | $1.76 \cdot 10^{-11}$ | $2.60 \cdot 10^{-2}$ |
| DER | $4.50 \cdot 10^{-2}$ | $9.29 \cdot 10^{-6}$ | $4.21 \cdot 10^{-6}$ | $2.07 \cdot 10^{-7}$ | $3.00 \cdot 10^{-3}$ |

## 6 Conclusion and Discussion

Instance-level continual learning addresses the challenging task of recognizing and remembering specific instances of object classes in an incremental setup, where new instances appear over time. This approach forms a more fine-grained challenge than conventional continual learning, which typically focuses on incremental discrimination at the class level. In this paper, we introduced a method for continually learning at both instance and class levels, arguing that real-world continual understanding requires recognizing samples at multiple layers of granularity. We observed that classes and instances form a hierarchical structure that can be leveraged to enhance learning at both levels. To this end, we proposed HyperCLIC, a hyperbolic continual algorithm designed for jointly learning instances and classes. We introduced continual hyperbolic classification and hyperbolic distillation, which embed the hierarchical relationships between classes and from classes to instances. Our experiments demonstrated that HyperCLIC operates effectively at multiple levels of granularity and achieves superior hierarchical generalization, consistently outperforming strong continual learning baselines. We conducted ablations on different distillation losses, backbone architectures, and pretrained models. Additionally, we presented qualitative analysis on class-level errors and provided evidence that our method makes less hierarchically severe mistakes. HyperCLIC enables real-world continual understanding, where recognizing and remembering both instances and classes over time is crucial.

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

# 7 Appendix

**Split CIFAR-100** In addition to evaluating our results on the EgoObjects dataset (Zhu et al., 2023), which serves as an instance-level continual learning benchmark, we also benchmark our method on iCIFAR-100 (Krizhevsky et al., 2009) to enable comparisons with other methods. Table 8 presents the performance of HyperCLIC and iCaRL (Rebuffi et al., 2017) on the iCIFAR-100 dataset under both pretrained and training-from-scratch conditions. When training from scratch, HyperCLIC demonstrates performance comparable to iCaRL in terms of instance- and class-level accuracy while surpassing iCaRL in LCA distance and super-class accuracy, indicating that the model effectively learns hierarchical representations. A lower LCA distance signifies that the model's errors adhere more closely to the underlying hierarchy. We also observe a higher degree of forgetting compared to the EgoObjects dataset. When pretrained, HyperCLIC consistently achieves superior hierarchical metrics than the baseline, although it still exhibits higher forgetting rates. Additionally, there is a noticeable difference between pretrained and from-scratch performance in EgoObjects versus iCIFAR-100, likely due to pretrained backbones being trained with class-level supervision that reduces intra-class variance.

Regarding forgetting, we note that HyperCLIC achieves higher classification accuracy on each new task, learning more than the baselines, though it also tends to forget more information. On the other hand, the baselines learn less on each new task but exhibit less forgetting. This trade-off between learning new information and forgetting is characteristic of methods that aim to build and refine hierarchical representations, as HyperCLIC does.

Table 8: **Results on the Split CIFAR-100 dataset** with a ResNet34 backbone. HyperCLIC consistently outperforms iCaRL in hierarchical metrics. At the instance level, HyperCLIC is comparable with iCaRL without pretraining and better with pretraining.

| | Pretrain | Accuracy ↑ | | | LCA ↓ | Forgetting ↓ |
| | | Instance | Class | Superclass | | |
|---|---|---|---|---|---|---|
| iCaRL | | 47.55 | 62.32 | 73.38 | 2.22 | 14.36 |
| HyperCLIC | | 45.52 | 62.31 | 74.76 | 2.15 | 24.76 |
| iCaRL | ✓ | 55.86 | 70.40 | 79.82 | 2.12 | 11.50 |
| HyperCLIC | ✓ | 56.46 | 74.11 | 84.52 | 1.95 | 15.39 |

**CORe50** We evaluate HyperCLIC and iCaRL on the CORe50 new class setting. As shown in Table 9, HyperCLIC outperforms iCaRL by 1% in instance-level accuracy, 4% in class-level accuracy, and 6% in superclass-level accuracy, all without requiring heavy hyperparameter tuning. It is important to note that CORe50 is a toy dataset with only 50 instances, making the hierarchy quite limited. Thus, leading to minimal differences between HyperCLIC and the baseline. However, as demonstrated in the EgoObjects dataset, our method excels in large-scale datasets with more complex hierarchies, where it can fully leverage its advantages.

Table 9: **Results on the CORe50 dataset at NC (new classes) scenario** with a ResNet34 backbone. HyperCLIC outperforms iCaRL in instance-level and hierarchical metrics.

| | Accuracy ↑ | | | LCA ↓ | Forgetting ↓ |
|---|---|---|---|---|---|
| | Instance | Class | Superclass | | |
| iCaRL | 21.41 | 32.45 | 41.71 | 3.16 | $-1.097 \cdot 10^{1}$ |
| HyperCLIC | 22.89 | 36.80 | 47.13 | 3.01 | $-1.159 \cdot 10^{1}$ |

**Extra Runs** In continual learning, the order of class presentation can significantly affect the final results. To demonstrate the generality of our method, we run HyperCLIC and the best-performing

baselines with 5 different random seeds in both from-scratch and pretrained settings on the EgoObjects benchmark, and report the average and standard deviations. Table 10 presents the results of this experiment. The findings suggest that HyperCLIC outperforms both DER and iCaRL in all metrics across both settings.

Table 10: **The average $\pm$ standard deviation of HyperCLIC , iCaRL, and DER, evaluated with 5 different random seeds in both from-scratch and pretrained settings on the EgoObjects benchmark.** HyperCLIC outperforms both strong baselines across all metrics.

| | Pretrain | Accuracy $\uparrow$ | | | LCA $\downarrow$ | Forgetting $\downarrow$ |
|---|---|---|---|---|---|---|
| | | Instance | Class | Superclass | | |
| iCaRL | | $21.49 \pm 0.91$ | $22.51 \pm 0.87$ | $23.62 \pm 0.85$ | $5.45 \pm 0.01$ | $8.72 \pm 1.15$ |
| DER | | $18.69 \pm 2.63$ | $19.89 \pm 2.58$ | $20.56 \pm 2.47$ | $5.46 \pm 0.10$ | $33.90 \pm 1.18$ |
| HyperCLIC | | $38.55 \pm 2.37$ | $42.74 \pm 2.50$ | $44.68 \pm 2.63$ | $4.91 \pm 0.03$ | $3.34 \pm 1.39$ |
| iCaRL | $\checkmark$ | $81.63 \pm 0.20$ | $87.03 \pm 0.15$ | $87.77 \pm 0.16$ | $3.91 \pm 0.01$ | $3.74 \pm 0.30$ |
| DER | $\checkmark$ | $80.73 \pm 0.59$ | $85.37 \pm 0.83$ | $86.13 \pm 0.80$ | $4.15 \pm 0.16$ | $9.98 \pm 0.53$ |
| HyperCLIC | $\checkmark$ | $83.14 \pm 1.24$ | $90.83 \pm 0.91$ | $91.75 \pm 0.81$ | $2.94 \pm 0.03$ | $5.61 \pm 1.5$ |

