# OpenReview forum: "Continual Hyperbolic Learning of Instances and Classes"
_TMLR — Rejected by TMLR_

### Review · Reviewer_4TRt · 2024-11-02

**Summary Of Contributions:**

This work introduces a novel approach for continual learning that enables simultaneous instance- and class-level incremental learning by embedding hierarchical relationships in hyperbolic space. This structure-aware method uses hyperbolic classification and distillation losses to retain task knowledge and reduce forgetting, achieving better performance over existing methods like iCaRL. Empirical results on benchmarks such as EgoObjects and CIFAR-100 demonstrate HyperCLIC’s advantages in accuracy, hierarchical consistency, and reduced forgetting, particularly when pre-trained backbones are used.

**Audience:**

Yes

**Claims And Evidence:**

No

**Requested Changes:**

1. What's the final loss for training the model? As several losses are introduced, but there is no description about how these losses are combined.
2. Implementation details of CIFAR-100 benchmark should be added in Sec. 4.
3. The experiments should be much more comprehensive as mentioned in weaknesses.

**Strengths And Weaknesses:**

The strength of the proposed method is hierarchical awareness. By embedding data in hyperbolic space, HyperCLIC captures the hierarchical relationships between classes and instances, which improves accuracy and reduces the impact of misclassifications.
The first half of the paper is quite well-written; however, the latter half does not convincingly demonstrate the effectiveness of the proposed method. Specifically, the experimental results lack generality. The compared methods are weak baselines, and only two benchmark datasets are tested. For the CIFAR-100 results in particular, only one baseline is compared. The fact that other baselines do not perform well on one benchmark does not necessarily indicate they will consistently perform worse on other benchmarks. Additionally, none of the reported results show statistical significance.

---

> ### Author Response · Authors · 2025-01-27
> **Response to reviewer 4TRt**
>
> We thank the reviewer for the positive feedback on the hierarchical awareness of our method and the writing of the first half of the paper. We have adopted the suggestions from the reviewer to improve the paper. Below, we address the raised points one by one.
>
> # Clarification of the loss function
> We apologize for any confusion regarding the losses in our method.
> Our method consists of two stages. The objective of the first stage (Section 3.1) is to obtain a set of hyperbolic prototypes, which are later used in the second stage (Section 3.2) for the classification loss. In the first stage, we sequentially apply three loss functions:
> 1. Poincaré Loss: Used for initializing the hyperbolic prototypes.
> 2. Entailment Loss: Enforces an entailment structure on the prototypes, ensuring that the hierarchical relationships are respected.
> 3. Separation Loss: Applied to address the issue of the entailment loss pushing leaf nodes too close to each other, ensuring proper separation between prototypes.
>
> This sequential application of losses follows the convention of gradient-based hyperbolic prototype learning methods.In the second stage, the prototypes obtained from the first stage are used in the classification loss, which is combined with the distillation loss to form the final loss function, similar to the approach in iCaRL.
>
> To further clarify the two-stage nature of our method, and following the reviewer’s suggestion, we have included two algorithm blocks in the revised manuscript. These blocks explicitly outline the workflow of each stage, providing a clearer understanding of how the losses are utilized.
> # Implementation details
>
> We thank the reviewer for pointing out the implementation details. We have added the implementation details for CIFAR-100 to Section 4 of the revised manuscript. Additionally, we have included the implementation details for the new benchmark (CORe50) to ensure completeness and reproducibility.
>
> # Benchmarks
> This paper is about instance-level continual learning. This is a highly challenging task with limited benchmarks. EgoObjects forms the only dataset with significant amounts of instances and classes, hence this dataset has our sole focus. We added a comparison to iCaRL for Split CIFAR-100 to show that our method works as expected for simpler datasets. There is unfortunately no other instance-level continual dataset with a non-trivial hierarchical structure. We did include results for Core50, which has only 50 instances across 10 classes, where we perform comparable to iCaRL. We have moved the CIFAR-100 and Core50 experiments to the appendix to keep the focus on EgoObjects as the only relevant benchmark. We see a great potential for additional large instance-level continual datasets due to the real-world relevance.
> The Table below shows the Core50 results:
> | Method   | Acc    | S      | C      | LCA    | Forgetting |
> |----------|--------|--------|--------|--------|------------|
> | iCaRL    | 21.41  | 32.45  | 41.71  | 3.1612 | -10.97     |
> | Ours     | 22.89  | 36.80  | 47.13  | 3.0160 | -11.59     |
>
> # Evaluating statistical significance
> We thank the reviewer for suggesting statistical testing to evaluate the generality and stability of our method across different runs. To address this, we selected the best-performing baselines, iCaRL and DER, and our method, in both from-scratch and pretrained settings. We conducted each experiment with 5 different random seeds and report the average and standard deviation of the results in Appendix Table 10.
>
> In addition, we performed statistical testing to compare HyperCLIC with the baselines. The table below presents the p-values for each metric. The results indicate that there are statistically significant differences between HyperCLIC and the two baselines across all metrics. Specifically, all p-values are below the commonly used threshold of p < 0.05, confirming that the observed performance improvements are statistically significant.
> |                   | From Scratch                     |                               | Pretrained                       |                               |
> |-------------------|-----------------------------------|-------------------------------|----------------------------------|-------------------------------|
> |                   | Acc          | S        | C        | LCA         | Forgetting | Acc         | S        | C        | LCA         | Forgetting |
> | iCaRL and HyperCLIC | 3.80 * $10^-7$ | 1.46 * $10^-7$ | 1.45 * $10^-7$ | 1.19 * $10^-9 $| 1.59 * $10^-4$ | 2.84 * $10^-2$ | 1.61 * $10^-5$ | 4.98 * $10^-6$ | 1.76 * $10^-11$ | 2.6 * $10^-2$ |
> | DER and HyperCLIC  | 1.53 * $10^-6$ | 5.79 * $10^-7$ | 3.96 * $10^-7$ | 2.71 * $10^-6$ | 2.84 * $10^-10$ | 4.5 * $10^-2$  | 9.29 * $10^-6$ | 4.21 * $10^-6$ | 2.07 * $10^-7$  | 3.00 * $10^-3$ |
>
> We hope that we have sufficiently addressed the reviewer's points.

---

### Review · Reviewer_yNK6 · 2024-11-25

**Summary Of Contributions:**

This paper tackles instance-level continual learning. To this end, they propose a hyperbolic continual learner that leverages the class-instance hierarchy for joint instance- and class-level recognition. Experimental evaluation on EgoObjects and Split CIFAR-100 highlight the potential of their approach.

**Audience:**

Yes

**Claims And Evidence:**

No

**Requested Changes:**

Please see the weakness.

**Strengths And Weaknesses:**

Strengths:
- Instance-level continual learning is interesting.
- The adaptation of hyperbolic geometry for this problem is also novel.

Weaknesses:
1. The superiority of hyperbolic space compared to Euclidean space is not well illustrated.
2. The only novelty lies in the adaption of hyperbolic space for this problem, which is somewhat limited.
3. Concern on experiment:
    1. For Table 1, why there is no comparison with the method after 2021?
    2. For Table 2, it is not convincing to only compare with iCaRL. Considering only two datasets in the experiment, more comparison is needed.  In addition, why your method achieves sizeable improvement on class and superclass, but limited improvement on instance? And your method is inferior in forgetting metric.

---

> ### Author Response · Authors · 2025-01-27
> **Response to reviewer yNK6**
>
> We thank the reviewer for the positive feedback on the adaptation of hyperbolic geometry for the problem of instance-level continual learning and suggestions to improve the paper. Below, we address the raised points.
>
> # Novelty
>
> We want to clarify the novelty of our work. First, we define a novel problem in continual learning that jointly addresses instance- and class-level continual learning, a challenge unexplored in prior work. Current literature only has instance-level awareness, but in many real-world settings such as robotics, a model needs to be aware of instances and classes simultaneously. This problem is hierarchical in nature and we demonstrate that hyperbolic space is a natural solution, as it effectively represents hierarchical relationships by embedding tree-like data with lower distortion compared to Euclidean space. Additionally, we propose a hyperbolic framework called HyperCLIC, establishing a foundation for hyperbolic hierarchical continual learning. Our empirical results demonstrate that HyperCLIC not only achieves superior instance-level accuracy but also preserves hierarchical consistency. We consider this work a foundational step toward advancing research in hyperbolic hierarchical continual learning.
>
> # Baselines
>
> For our comparisons, we focus on the most canonical and well-known approaches in continual learning, such as iCaRL and DER. Many recent continual learning methods integrate these approaches with vision-language models and transformers [1,2]. However, comparing directly to such baselines would not provide an apples-to-apples comparison, as our method uses a ResNet backbone and does not rely on pretrained vision-language models. We believe this choice ensures a fair and focused evaluation. That said, HyperCLIC is compatible with transformer-based architectures and VLMs, which we consider an exciting avenue for future research. Please let us know if there are any baselines from recent literature that are desired.
>
>
> # Additional baselines for Table 2
>
> Our paper is about instance-level continual learning, with EgoObjects as the only large challenging benchmark. We included a Split CIFAR-100 experiment to show that our approach is not strictly limited to EgoObjects. As pointed out by reviewer Gzjj, CIFAR-100 is hierarchically limited. It is also not instance-level but class-level. To avoid confusion, we have moved the Split CIFAR-100 experiments to the appendix. We strongly believe that our hyperbolic approach is the way to go for joint instance- and class-level continual learning, as demonstrated on EgoObjects. Unfortunately, no other similar benchmark exists in current literature.
>
> # On forgetting and accuracies
> Regarding the performance discrepancy between class-level, superclass-level, and instance-level accuracy, our method enforces a hierarchical structure, which contrasts with Euclidean-based methods like iCaRL and DER. We consider the deeper improvements at the class- and superclass-level a bonus and a strength of our method. Since we consider the full hierarchy, we benefit at all stages.
> Regarding forgetting, we note that HyperCLIC achieves higher classification accuracy on each new task, learning more than the baselines, though it also tends to forget more information. On the other hand, the baselines learn less on each new task but exhibit less forgetting. This trade-off between learning new information and forgetting is characteristic of methods that aim to build and refine hierarchical representations, as HyperCLIC does.
> We hope that our clarifications and additional experiments have addressed the reviewer’s points.
>
> [1] Douillard, Arthur, et al. "Dytox: Transformers for continual learning with dynamic token expansion." Proceedings of the IEEE/CVF Conference on Computer Vision and Pattern Recognition. 2022.
>
> [2] Wang, Yabin, Zhiwu Huang, and Xiaopeng Hong. "S-prompts learning with pre-trained transformers: An occam’s razor for domain incremental learning." Advances in Neural Information Processing Systems 35 (2022): 5682-5695.

---

### Review · Reviewer_Gzjj · 2024-12-23

**Summary Of Contributions:**

This paper addresses instance-level continual learning, a challenging task in an incremental learning setup that involves the recognition and classification of new instances. The authors propose a method for continual learning at both instance and class levels, emphasizing that real-world continual learning understanding requires recognition at both levels of granularity. The authors first observed that classes and instances form a hierarchical structure, which can be leveraged to enhance learning at both levels. To this end, they introduced HyperCLIC, a hyperbolic continual learning algorithm designed for joint learning of instances and classes. The method incorporates hyperbolic continual classification and hyperbolic distillation. Their experimental results demonstrated that HyperCLIC operated effectively across multiple granularity levels, achieving superior continual learning performances on two benchmark continual learning datasets.

**Audience:**

Yes

**Claims And Evidence:**

Yes

**Requested Changes:**

1. Regarding the second weakness, could the author conduct additional experiments using other benchmark datasets that contain a hierarchy between instances and classes?

2. The method seems required to tune its hyperparameters finely. Can you explain how many hyperparameters the method has and experimentally show how sensitive the method's performance is by varying each hyperparameter?

3. It seems unclear to me if the two learning objectives in Sections 3.1 and 3.2 are used at the same time or optimized alternatively for each task and if each instance in one task is coming sequentially or together. Adding a algorithm illustration for the whole process of the method could be helpful for clearer explanation.

**Strengths And Weaknesses:**

* Strengths
1. This paper shows a new method that utilizes hyperbolic learning for continual learning to leverage hierarchical information for a given task.
2.  This paper demonstrates that the proposed method outperforms continual learning performance on two benchmark datasets.

* Weaknesses
1. The learning objective function of the proposed method contains several terms where we should tune hyperparameters to balance each term. Furthermore, it also contains other hyperparameters such as a temperature scale we have to tune to get appropriate hyperbolic embedding features.

2. The method was evaluated on only two benchmark datasets. Because iCIFAR-100 doesn't have any hierarchical structure between instances and classes, additional experiments with different datasets would be needed to check if the method works well in other settings.

---

> ### Author Response · Authors · 2025-01-27
> **Response to reviewer Gzjj**
>
> We thank the reviewer for the positive feedback on the novelty and the performance of our method. Below, we address the raised points by the reviewer one by one. Note that three tables have been omitted from this response due to character shortage. You can find them in the paper.
>
> # Additional benchmark
> This paper is about instance-level continual learning. This is a highly challenging task with limited benchmarks. EgoObjects forms the only dataset with significant amounts of instances and classes, hence this dataset has our sole focus. We added a comparison to iCaRL for Split CIFAR-100 to show that our method works as expected for simpler datasets. There is unfortunately no other instance-level continual dataset with a non-trivial hierarchical structure. We did include results for Core50, which has only 50 instances across 10 classes, where we perform comparable to iCaRL. We have moved the CIFAR-100 and Core50 experiments to the appendix to keep the focus on EgoObjects as the only relevant benchmark. We see a great potential for additional large instance-level continual datasets due to the real-world relevance.
> The Table below shows the Core50 results:
> |Method|Acc|S|C|LCA|Forg|
> |-|-|-|-|-|-|
> |iCaRL|21.41| 32.45  | 41.71|3.16|-10.97|
> |Ours| 22.89  | 36.80| 47.13| 3.01|-11.59|
> # Additional ablations
> We agree with the reviewer that understanding the sensitivity of the method to its hyperparameters is crucial for evaluating its robustness and usability. To address this, we propose a two-stage approach. In the first stage, we focus on building high-quality prototypes. In the second stage, we combine these hyperbolic prototypes with continual learning through classification and distillation losses.
> For the first stage, we conducted an ablation study on the most critical hyperparameters that define the quality of the prototypes: the hyperbolic embedding dimension and the number of entailment epochs. The results, presented in Table 2, show that a very small embedding dimension fails to capture an effective hyperbolic representation. However, performance improves with medium-sized and larger dimensions, which are competitive. Additionally, we observed that increasing the number of entailment epochs negatively impacts performance. This occurs because the entailment loss pushes the children of the same parent too close to each other, leading to a deterioration in both instance-level accuracy and hierarchical metrics.
>
> For the second stage, the key hyperparameters are $\tau$ and $\lambda$. The temperature influences the classification loss by adjusting the smoothness of the probability distribution derived from the pairwise distances of the hyperbolic prototypes and is typically set to 0.1 in hyperbolic literature (Ibrahimi, et al 2024). Our results, shown Table 3, demonstrate that HyperCLIC is sensitive to temperature, particularly when trained from scratch, as compared to the pretrained model.
>
> The continual learning objective function in the second stage of our proposed method combines classification and distillation losses, following the approach in iCaRL. In this experiment, we balance these losses using \lambda to evaluate the influence of each loss term, as shown in the following loss function: $Loss =  \lambda * L_{distillation} + (1-\lambda) * L_{classification}$. Our results in Table 5 suggest that a fully balanced loss yields the best performance in HyperCLIC.
>
> Overall, we conclude that our approach is robust to various hyperparameter settings. All analyses have been added to the manuscript.
>
> # Clarification of the loss function
> Our method consists of two stages. The objective of the first stage (Section 3.1) is to obtain a set of hyperbolic prototypes, which are later used in the second stage (Section 3.2) for the classification loss. In the first stage, we sequentially apply three loss functions:
> 1. Poincaré Loss: Used for initializing the hyperbolic prototypes.
> 2. Entailment Loss: Enforces an entailment structure on the prototypes, ensuring that the hierarchical relationships are respected.
> 3. Separation Loss: Applied to address the issue of the entailment loss pushing leaf nodes too close to each other, ensuring proper separation between prototypes.
>
> This sequential application of losses follows the convention of gradient-based hyperbolic prototype learning methods.In the second stage, the prototypes obtained from the first stage are used in the classification loss, which is combined with the distillation loss to form the final loss function, similar to the approach in iCaRL.
> To further clarify the two-stage nature of our method, and following the reviewer’s suggestion, we have included two algorithm blocks in the revised manuscript. These blocks explicitly outline the workflow of each stage, providing a clearer understanding of how the losses are utilized.
>
> We hope that the clarifications, new analyses, and additional clarifications to the method have addressed the reviewer’s points.

---

### Author Response · Authors · 2025-01-27
**General answer**

# General answer


We thank the reviewers for their suggestions to improve the paper and the editor for granting the time to make these improvements. Overall, the paper has been extended along the following dimensions:

- **Ablations.** Following the reviewers’ guidance, additional ablations on the hyperbolic embedding dimension, entailment epochs, temperature and loss balancing hyperparameters have been included.
- **Method clarity.** We have expanded the explanation of our method and we have added 2 algorithm blocks to outline the steps involved in our approach.
- **Statistical analysis.** We have rerun the most competitive methods on EgoObjects for 5 runs, allowing us to show that our method obtains statistically significant improvements.
- **Clarification on dataset choice.** Our focus is instance-level continual learning. For this research problem, EgoObjects is the only large-scale dataset. The Core50 dataset is also an instance-level dataset, but is a toy dataset with only 50 instances, resulting in minimal differences between methods. We have added Core50 with a comparison to iCaRL (the closest Euclidean counterpart to our method) to Table 9. We strongly believe in the match between hyperbolic geometry and joint class- and instance-level continual learning, but there is unfortunately a lack of large instance-level continual datasets.

The 3 extra pages, 6 tables, 2 algorithms, 39 experiments have been added to the paper. All changes are highlighted in blue in the updated manuscript. We will address all reviewer comments in detail in the individual responses below.

---

### Decision · Action_Editor_GtEs · 2025-02-22

**Recommendation:** Reject

**Comment:**

The paper received mixed reviews. The reviewers acknowledged that the work proposes a novel continual learning method that leverages hierarchical relationships between instances and classes, demonstrating good performance on multiple benchmarks. However, they raised several major concerns on the initial submission:
1. Insufficient experimental evaluation (Gzjj, 4TRt): The evaluation was conducted only on two benchmark datasets, which restricts the generality of the conclusion.
2. Missing recent baseline methods in comparison (yNK6, 4TRt): The experiments did not include recent class incremental learning methods (e.g., those published after 2021), and for some comparisons, only one or two baseline methods (e.g., iCaRL) were included. The small sample size also lacked statistical significance.
3. Lack of clarity in training losses and procedure (Gzjj, 4TRt): It was unclear how the multiple losses are used and combined during model training.
4. Lack of sensitivity analysis on hyperparameters and performance analysis across hierarchy levels (Gzjj, yNK6).

In their response, the authors partially addressed these concerns by (1) adding results on an additional benchmark CORe50, (2) clarifying the choice of baselines and its training losses, (3) providing sensitivity analysis of hyperparameters and statistical significance. After the rebuttal, while the reviewers found their concerns about points 1, 3 and 4 largely addressed, the limitation regarding the scope of experimental comparison remained unresolved. In particular, Reviewer 4TRt was unconvinced by the insufficient baselines and incomplete comparisons, feeling that the claims were not well supported. Reviewer yNK6 also emphasized the need for more comprehensive comparisons.

The AE finds the argument of Reviewer 4TRt and yNK5 more persuasive. The paper only considered ResNet-based classifiers and the corresponding class incremental learning (CIL) methods, which is limited by current standards. In addition, more comprehensive comparisons with different baselines for each benchmark are missing in the current version. These limitations weaken the evidence supporting the claims and does not convincingly demonstrate the contributions and performance improvements of the proposed approach. As a result, the AE regrettably recommends rejection. The authors are encouraged to revise the paper by addressing these limitations and consider resubmitting at a later time.

**Audience:**

The topic of continual learning is of interest to general machine learning audience.

**Claims And Evidence:**

The paper proposes a hyperbolic continual learning strategy that incrementally learns both classes and instances simultaneously. It leverages hyperbolic representations to encode the class-instance hierarchy and introduces hyperbolic classification and distillation losses to perform incremental learning of instances and classes.

The authors present experimental results showing improved performance on three benchmarks, EgoObjects, Split CIFAR-100, and CORe50, compared to previous methods such as iCaRL and DER. While these results suggest potential advantages of the proposed approach, the reviewers raised concerns about the experimental evaluation regarding the limited scope in comparisons, including lack of recent baselines and incomplete comparisons to previous methods. Due to these limitations, the evaluation does not convincingly demonstrate the effectiveness and superiority of the proposed strategy as claimed.

**Resubmission Of Major Revision:**

The authors may consider submitting a major revision at a later time.